# Building Resilience into Classrooms: A Participatory Action Approach

Beth Doll * and Kyle Bizal

Department of Educational Psychology, University of Nebraska Lincoln, Lincoln, NE 68588, USA;
ebizal2@huskers.unl.edu
* Correspondence: bdoll2@unl.edu

**Abstract:** The ClassMaps strategy builds resilience into classrooms by translating the compelling results of developmental risk and resilience research into simple action plans that embed positive protective supports into classroom routines and practices. The refined routines reinforce students' developing self-agency and foster their nurturing relationships with adults and peers. The strategy is carried out in four stages—administering the ClassMaps Survey to describe students' experiences of classroom strengths and weaknesses, conducting a classroom meeting with students to scrutinize the results and brainstorm plans for change, implementing simple modifications to classroom routines and practices based on that plan, and evaluating the impact of the modifications a few weeks later. The ClassMaps Survey's central role in the strategy is justified by the reliability of its subscales and their relation to the key protective supports identified in resilience research. The ClassMaps strategy is a useful example of 'giving psychology away'.

**Keywords:** resilience; peer relationships; teacher–student relationships; home–school relationships; self-determination; self-control; self-efficacy; classroom climate

## 1. Introduction

Fifty years of developmental risk and resilience research has established that children's psychological well-being and their capacity to overcome adversity is closely related to the availability of protective supports from their families, communities, and schools [1]. Compelling examples of these studies include Masten et al.'s [2] longitudinal study of children in Minneapolis public housing; Werner and Smith's [3] examples of bounce-back children; and Rutter et al.'s [4] Isle of Wight studies. Initially designed as longitudinal studies of the developmental impact of adversity, the researchers identified cohorts of young children and carefully tracked their exposure to developmental risks such as poverty, child maltreatment, the poor health of the child or their parents, or parental discord. They had originally anticipated that different forms of adversity would lead to different forms of adult dysfunction, and that the severity of the adversity would predict the likelihood that a child would succumb to risk. Unexpectedly, the results showed that the *number* of adversities that children encountered was a critical variable: children growing up with at least three or four important life adversities were far more vulnerable to multiple alternative forms of maladaptive adult outcomes. Even more striking, as they followed their cohorts into adulthood, each researcher realized that their study included a subset of children who were growing up with a significant number of life adversities and so should have been failing but were nevertheless developing into successful adults. Upon revisiting their longitudinal data, a common finding from multiple researchers was that these vulnerable children who became successful despite the odds had benefited from multiple positive, protective supports [1]. While some of these protective supports were characteristics of the children themselves (peer friendships, self-efficacy, self-determination), others were instead characteristics of their families and communities (close bonds with nurturing

adults, access to prosocial community organizations, effective parenting, and effective schools). Like the relationship between adversity and adult outcomes, it was not the *nature* but the *sheer number* of positive protective supports that predicted children's capacity to overcome adversity. The studies also showed that schools played a particularly important protective role in children's resilience. Rutter and his colleagues [5] noted that school factors only predicted 20% of the variance in children's ultimate life success, but these school characteristics were the most malleable factors and were most prone to change in ways that predicted the children's success.

The compelling results of this developmental risk and resilience research can be summarized in three simple sentences: Risk is often imposed on children by an adult world that has not protected them from harm. When this occurs, the availability of multiple positive caretaker and community supports are powerful predictors of the children's ultimate resilience in the face of risk. And numbers count; the more protective factors that are embedded into children's daily lives, the greater the chances that children will subsequently overcome adversity to succeed as adults.

These lessons hold important implications for schools [1]. First, schools must do no harm. It is essential that schools identify and neutralize children's exposure to social adversities to the degree that they are able, and particularly during the times that children spend in school. Next, schools must amplify the number and strength of protective supports available to children during their time in school. Schools have the capacity to embed protective supports into students' daily routines: peer friendships, personal experiences of success, caring adult interactions, and opportunities to make decisions and build on personal strengths during the school day. In essence, this is a population-based charge to the school—to ensure that every enrolled child is nurtured repeatedly and in multiple alternative ways. Within the framework of Bronfenbrenner's [6] classic model of ecological development, developmental risk and resilience research suggests that school-based strategies to promote children's wellbeing might sometimes work in the outer rings of students' ecosystems—modifying the routines and practices in classrooms and schools—as a strategy to protect students' social and emotional well-being.

A striking example of the power of the ecosystem occurred during a snowy Denver, Colorado, recess. The first author and several graduate students were systematically observing six elementary students from an intervention group for children without friends. On a typical day, they played alone for most or all observation intervals. The children had assorted explanations for their isolation. As an example, one 4th grader explained that he was a strong proponent of nonviolence, like Mahatma Ghandi, and the other students on his playground were too violent to play with. On this snowy day, however, the recess game was rolling snowballs—four-foot and five-foot snowballs that required large teams of students to push them around the playground. Remarkably, every one of the friendship group children were included in one of the snowball teams. They were all laughing and talking with other students, and they had a great time. It became clear that changing the game on the playground was far more effective than the friendship group in fostering these children's peer relationships. This focal event shifted our thinking about how to translate developmental risk and resilience research into schools and classrooms.

This paper will describe one such translation: the ClassMaps strategy for making classrooms and schools resilient [7]. The book's title of *Resilient Classrooms* was not without its critics; one reviewer commented to the publisher that the title was a mistake, because "everyone knows that children are resilient, not schools". This paper contends that settings (schools, classrooms, playgrounds) can and should be refined to predispose children to resilience. This paper will clarify the underlying rationale to our resilient classrooms strategy and will provide a brief updated summary of the empirical research related to the ClassMaps Survey (CMS) that is central to this strategy.

## 2. The ClassMaps Strategy

The ClassMaps strategy described here emerged out of an attempt to find and implement 'snowball games'—school practices that embed protective factors into students' every day. The ultimate goal is to make schools into protective environments that predispose children to be academically, socially, and behaviorally successful despite the adversities they may be experiencing elsewhere in their lives. This strategy translates developmental risk and resilience research into six characteristics that could be embedded into classroom learning environments:

a. academic efficacy: students expect to be successful and act in way that makes their success more likely;

b. academic self-determination: students have personal goals for their own learning and development and are able to translate these goals into actions;

c. behavioral self-control: students self-regulate their own conduct to promote their success;

d. teacher–student relationships: students have relationships with their teacher that are warm, authentic, responsive, and predictable;

e. effective peer relationships: students have supportive friends in class and are able to resolve occasional conflicts without disrupting their friendships;

f. close home–school relationships: students and their parents talk about the students' school experiences, including both their successes and their challenges.

The intent is to recognize these as characteristics of the system of the classroom and not simply of the children themselves.

Over the past two decades, substantial evidence has accrued to show that these classroom experiences are predictive of the development of students' executive functioning [8], their academic engagement [9,10], and their capacity to succeed despite substantial lifetime adversity [2]. The three self-agency characteristics of classrooms support students' self-direction of their own learning. Students' growing academic efficacy fuels their commitment to learning [11]. When they expect to succeed at any academic task, they become more focused, more persistent, and more attentive; and if they are not immediately successful, the efficacious student will change strategies, try harder, or work longer. This is related to, but different from, students' academic self-determination. Self-determined students have personal goals for their learning, systematically plan for ways to reach those goals, and act on their plans [12]. They take charge of their learning. Students' direction of their own learning requires that they be self-controlled—that they can manage their moment-to-moment behavior to align with their goals. Inevitably, classroom routines and practices that foster students' efficacious, self-determined learning strengthen their engagement with academic tasks and increase their experiences of success.

The three relational characteristics of classrooms foster students' sense of belongingness at school, provide the moments of joy that associate school with happiness, and ultimately support higher attendance and more focused attention to learning. When students experience their teachers as warm and responsive, they feel respected, enjoy learning, and are more persistent when tasks are challenging. Similarly, students who have friends in class feel emotionally safe, can ask for assistance if a task is hard, and enjoy the daily activities of the classroom [13]. Home–school relationships represent the juxtaposition of students' two most important sources of support—their families and their schools. When students believe that their families know about and value their school experiences, they participate more actively in classroom learning [10,14].

The ClassMaps strategy was developed with the essential understanding that school-based strategies to foster children's social and psychological well-being must be both practical and relevant to the core responsibility of schools. To be practical, the strategy needs to be time-efficient, available at little or no cost to the school, and embedded into the ongoing routines and practices of schools. This strategy focuses on embedded routines because there is limited room in most school curricula for extensive direct instruction on social and emotional competencies. Central to the ClassMaps strategy is a simple assessment of

the six resilience characteristics that is brief, easily administered, could be used to identify which resilience characteristics require improvement in a classroom, and could determine whether strategies to enhance these have been effective. The strategy is modeled after a familiar response-to-intervention framework that uses five-minute measures of reading, mathematics, and writing to systematically refine academic interventions [15]. To replicate this framework in a resilience-promoting intervention, the ClassMaps strategy needed five-minute measures of its protective factors to monitor and refine classroom routines and practices.

This need prompted the development of the ClassMaps Survey (CMS) [16,17]. Three alternative measurement strategies were considered: direct observation, teacher reports, and student reports of each school characteristic. While reliable and valid direct observations of the six classroom resilience characteristics were possible, these required extraordinary amounts of time to conduct reliably. Teacher reports were also reliable and valid measures of the classroom characteristics, but it was quickly apparent that teachers were not interested in completing surveys that were then collated and returned to them as graphs or reports. However, teachers were very intrigued by student surveys, as these provided them with information that they did not already possess. Ultimately, it became clear that student surveys could play a central role in shaping teachers' and students' reflections about their classrooms and opportunities for classroom change. To ensure that these reflections were rich and relevant, the content of the CMS items was drawn from the extensive research on developmental risk and resilience and its identified protective factors.

Through a cyclical, collaborative process, teachers carry out the ClassMaps strategy in four stages:

1. They administer the anonymous class-wide survey to collect students' perceptions of their classroom (the CMS);
2. They make sense of the survey results in partnership with colleagues and by drawing upon the collective experiences of their students through a classroom meeting;
3. They use these perceptions to plan and implement simple changes to classroom routines and practices that can improve students' classroom experiences;
4. They reassess the impact of those changes by re-collecting one or two CMS subscales and re-sharing these results with the students.

Some teachers elect to cycle through these stages more than once to repeatedly refine and strengthen the six protective factors in their classroom. Teachers elect to use the ClassMaps strategy because they are concerned about the behavior, relationships, routines, or attitudes that interfere with their students' success. The strategy depends on teachers' voluntary participation and is *never* used for teacher evaluation.

### 2.1. Administer the Anonymous CMS

Work on the ClassMaps strategy begins with students' anonymous completion of the CMS to describe their collective experiences of their classroom. Having been refined through a decade of research [16,17], the CMS has eight 5–8 item subscales representing the six characteristics of resilient classrooms. (Three of the subscales describe students' peer relationships, with one subscale each for the remaining classroom characteristics; see Appendix A for a copy of the CMS.) Students describe their level of agreement with the survey items using a four-point Likert-type scale from "Never" to "Almost Always". Although originally developed as a paper survey, most teachers now administer the CMS virtually using online survey software; all students in a class can complete it in less than 15 min. The anonymity of the CMS is important because it allows students and teachers to be frank in describing their own experience of the classroom. To further encourage honest responses, students are told in advance that they will see the results and will help plan ways to improve the classroom routines. In most cases, though, students do not believe that they will be true partners in the ClassMaps strategy until they experience it. As one seventh grader asked, "Is this all about fixing our school? Because, if we knew that before we answered the questions, I think we would tell the truth". His question poignantly

emphasized the importance of transparency and participatory planning for the ClassMaps strategy.

### 2.2. Make Sense of the CMS Results

In the second stage of the ClassMaps strategy, students' CMS responses are organized into eight student-friendly bar graphs that provide a "snapshot" of the classroom experience as reported by the students' responses (see Figure A1 for an example). Separate graphs are created for each subscale using a set of stacked bars, one bar for each subscale item. Colors make the graphs very intuitive to understand: red represents the number of very negative responses, green represents the number of very positive responses, and shades of blue or yellow represent the middle two responses. Lots of green on a graph describes something to celebrate, while lots of red signals 'pay attention'. Because the results may be widely shared with students and colleagues, no student or teacher is ever identified in these graphs.

Making sense of the results begins when the classroom teacher, alone or with a colleague, reviews all eight graphs to prioritize a graph identifying a classroom strength and another describing a classroom weakness that might be addressed. These selections are made by 'teacher judgment' rather than any score-based standard, acknowledging teachers as the experts in how their classrooms ought to function. Inevitably, some graphs will speak to the teacher because they surprise, reinforce, or clarify the teachers' prior insights about the class. Teachers may also select a graph because the behavioral data, grades, or observations also suggest that the weakness is a problem in the classroom. Graphs representing strengths give insight into what is working well; teachers can share these graphs with students to celebrate their successes and, perhaps, to leverage these strengths to address a weakness.

### 2.3. Convene a Classroom Meeting with Students

Teacher-led classroom meetings ensure that student insights are part of the planning for classroom change. Gathering student insights also establishes students' shared responsibility for and collective commitment to meaningful changes in the classroom. First, the teacher projects a graph describing a classroom strength to the full class and leads students through a brief exercise in reading the graph. Then, the teacher asks, "Is this really true about our class?" while a colleague or assistant keeps a running list of the students' comments on a chart tablet at the front of the room. A second question is asked about the strengths graph, "Why do you think that this happens in our class?" And, again, chart notes are kept of the various student comments.

As discussion of the strength comes to a natural conclusion, the teacher projects a graph describing a classroom weakness for the students. Four questions engage the students in understanding and planning for change: Is this really true about our class? Why do you think this happens in our class? What could adults do to make things better? And what could students do to make things better? This question sequence allows students to comment on the accuracy of the results, suggest hypotheses for the cause of the reported weaknesses, and suggest strategies for change. In most classes, students are quick to offer suggestions about what the teacher could do to address the problem, and once these comments come to a close, they are sometimes able to talk about what they and their classmates might do. These last suggestions are particularly important because these prompt students to take ownership of classroom changes.

### 2.4. Planning for Change

After the classroom meeting, the teacher has four important resources for planning changes in classroom routines or practices: (1) their own experiences of the classroom; (2) the graphs of the ClassMaps Survey results; (3) suggestions made by colleagues; and (4) the students' suggestions. Inevitably, teachers pick out three or four key student comments that are particularly insightful and that shift their working hypothesis for why a

problem is occurring and what to do about it. As an additional support for planning change, the 2nd edition of *Resilient Classrooms* [7] includes copy-ready strategy sheets linked to each CMS subscale. These describe actions taken by other teachers to address classroom weaknesses. An effective practice is to simply lay the strategy sheets out on a table as resources and allow teachers to pick among them as they find useful. Drawing from all sources, the teacher identifies a few simple modifications to classroom routines and practices that might address the weakness.

The critical word is 'simple'. Consistent with the overriding practicality of the ClassMaps strategy, modified routines must be viable, cannot require any significant amount of teacher time, should not interrupt or replace the instruction and learning that occurs in the classroom, and should not require significant additional resources that are not readily available to the class. The best plans are systemic, engaging the class as a whole rather than focusing on individual students. Effective plans for change include descriptions of who will carry out the new routines and when and how the teacher can judge that the plan is actually being put into practice. Because these changed routines grew out of teachers' conversations with students, they are ecologically valid and are likely to persist over time. Teachers nevertheless retain their authority over the class—they choose which graphs to share, choose which student insights to react to, and plan changes in routines and practices that fit their classroom.

Importantly, the people with the most time to contribute to the modified routines are the classroom's students. Because students tend to have more unscheduled time than teachers, they may be tasked with leading some routines and monitoring progress. For example, in a classroom addressing work completion, students were charged with implementing a checklist for seat work and self-monitoring task completion. In addition to saving the teacher's time, sharing the workload empowered these students to take charge of their learning spaces while fostering their self-reliance and autonomy.

*2.5. Evaluate and Refine the Plan*

Within two or three weeks, the teacher evaluates in order to decide whether their ClassMaps plan is working. The simplest way to do this is to re-administer the CMS subscale that originally prompted the plan for classroom change. Alternatively, some classrooms may have existing classroom records that could serve as valuable indictors of progress (e.g., grades, office referrals, or homework completion). As a third alternative, a second classroom meeting could be held to include students in the discussion of progress. Any plan that does not appear to be working, or has only some parts that are working, can be revised or replaced. New classroom routines that have been effective are embedded into the everyday practices of the classroom.

## 3. Examples of Actual ClassMaps Strategies That Teachers Have Led

Using the ClassMaps strategy, a seventh-grade middle school team learned during a classroom meeting that their students were receiving repeated suspensions over the lunchtime break. The students explained that it was because hanging out after lunch was so very boring. The teachers' ClassMaps plan was to send a dozen alternative games outside with the students after they had finished eating: hacky sacks, frisbees, a gigantic checkerboard, hula hoops, and jump ropes. The number of suspensions plummeted and, in this economically challenged community, the number of 7th graders dropping out of school also declined.

In their ClassMaps classroom meeting, a second-grade class explained that they were often disruptive because they did not notice when it was time to concentrate and get down to work [18]. The teacher and her students created a set of signals that they could use to remind each other that it was working time. A subsequent administration of the survey documented their improved attention.

During their classroom meeting, a fourth-grade class explained that their lunch time soccer games never actually got started because they struggled to agree on the teams. They

were angry and arguing in the afternoon class as the after-lunch soccer conflicts bled over into class time. As their strategy, the class decided to choose soccer teams for the week on each Monday, wrote standard procedures for choosing teams and refereeing the games, and modeled these after the international mediation strategies they were learning about in history. The new procedures gave them more time actually playing soccer and calmed down their afternoon work time. They developed their refined procedures into a manual and a formal presentation that they made to the other grades in their school.

Second graders in a classroom meeting explained that they argued constantly over the rules of all of their recess games—tetherball, soccer, four-square. When asked what adults could do to fix it, one student suggested, "Someone should teach us the rules". The teachers put this suggestion into action: the school's physical education teacher wrote out elementary-grade rules for the games, taught them in PE class, shared the same rules with the second-grade teachers and playground supervisors, and eventually established district-wide rules (because students living in rental housing moved frequently among schools in this district.) A second administration of the CMS, 6 weeks later, showed that playground arguments were much less frequent and no longer intruded into the classroom.

A defining strength of the ClassMaps strategy is that it is led by the classroom teacher and also emphasizes the insights and experiences of the students in the class. As classroom leaders, teachers are pivotal change agents with the authority to implement classroom improvements. As active classroom participants, students can reliably describe the experience of learning in their classroom and their insights catalyze the possibilities for classroom change. This simple four-step strategy is time efficient, data-based, and cost-effective. The CMS is open source, easily administered using widely available platforms (paper, Goggle docs, Qualtrics). The brevity of each phase makes it easy for a teacher to introduce the ClassMaps strategy into a class. The CMS provides data that are local to the classroom, prompt useful discussions of the students' experiences, and raise new possibilities for evidence-supported protective factors.

## 4. The ClassMaps Survey

Most of the empirical research related to the ClassMaps strategy has been an examination of the technical properties of the ClassMaps Survey (CMS). The CMS is an anonymous student survey with 55 items describing six characteristics of classroom learning environments derived from developmental risk and resilience research. CMS items emphasize classroom characteristics that have been repeatedly demonstrated in educational research to be related to student success [7]. These lines of research converged to target three aspects of students' classroom relationships (teacher–student relationships, peer relationships [friendships and conflict], and home–school relationships) and three aspects of students' developing autonomy (academic self-efficacy, academic self-determination, and behavioral self-control).

The CMS items use a simple four-point Likert response format: *Never*, *Sometimes*, *Often*, and *Almost Always*. Two comprehensive examinations assessed the initial validity and reliability of the CMS. The results from a sample of 345 third- through fifth-grade public school students showed that 53 of the 55 items loaded onto their predicted subscale, grade and gender effects were small, and the subscales were internally consistent, with coefficient alphas between 0.78 and 0.80 [16]. The analysis of a second sample of 1019 middle school science students again showed that the survey's items factored into their predicted subscales, with subscale coefficient alphas ranging from 0.82 to 0.91 [17]. A more recent examination of 542 elementary students and 517 middle school students in a northeastern USA school district showed that all items loaded onto their respective subscales, exhibited adequate model fit, and had good internal consistency ($\alpha = 0.78$ to 0.92) [19]. Interestingly, the CMS data from this northeastern school district were collected before, at the height of, and in the waning days of the COVID pandemic. The results showed that students' academic efficacy and academic self-determination declined during the pandemic, when students were socially distanced at school, and that these recovered once more typical

schooling resumed. Similarly, their relationships with teachers were less positive at the height of the pandemic but, for the middle grades, improved again once they were at school in person. Alternatively, the quality of students' peer friendships became more positive at the height of the pandemic and had stayed high three years later.

The CMS has been used as a measure of students' classroom experiences in independent studies, assessing the technical properties of the CMS in English and as translated into Chinese, Indonesian, and Urdu [20–23]. Factor analytic studies have confirmed that the items loaded consistently onto their assigned scale for both English and translated measures. Correlations were reported of the CMS with the Yale School Climate Survey [24] and the Protective Peers scale [25]. In two dissertations, Chapla [26] established that teachers were not able to predict in advance what their students subsequently reported on the CMS. Franta [27] demonstrated that, by fourth grade, almost all students could independently read and understand the CMS items.

The CMS provides a template of the school factors underlying resilience. Its items are derived from developmental risk and resilience research. It is brief, affordable, and appropriately readable for elementary and middle school students. It is easily adapted to student or teacher needs. The questions are not highly intrusive. Collectively, research from the past 15 years suggests that the CMS's technical properties are reasonable as a measure of students' classroom experiences, and that the CMS subscales can be used together or separately as very brief probes that can be used to track changes in students' experiences over time. The process of "survey → class meeting → plan for change → modify routines and practices" could be completed with other measures in place of the CMS. The alternative would need to be very brief, technically sound, and linked closely to the research on school resilience.

## 5. Key Attributes of the ClassMaps Strategy

*A Participatory Action Strategy*

At its core, the ClassMaps strategy is a 'participatory action' method for strengthening classroom learning environments. The term was first used by Lewin [28] when he suggested that participatory action research could be a continuous self-reflective cycle of inquiry, action, and evaluation conducted collectively with the members of the community that was the object of change. Within the ClassMaps strategy, teachers and their students work together in an iterative cycle of change to more fully understand their classroom and change it for the better. The administration of the anonymous ClassMaps Survey (CMS) initiates this iterative cycle of evaluation using students' collective experiences of learning in the classroom. The classroom meeting engages students and the teacher in careful reflection about the CMS results and the implications these hold for classroom change. The resulting action plan is co-constructed out of the conversation around student insights and teacher experiences. Importantly, the resulting action plan is 'low stakes'—if the readministered CMS suggests that the plan is not working well, the teacher and students can always refine and upgrade it. Engaging students as full participants in planning and carrying out the ClassMaps strategy gives them greater awareness of their situation in the classroom and builds their agency over the very conditions of their own learning.

## 6. Readily Adaptable to Technology

Early work using the CMS was conducted with paper surveys, and results were hand-entered into a spreadsheet for analysis. However, the CMS's brevity and practicality lends itself well to the use of technology to simplify survey administration, analysis, graphing, and presentation. In classrooms where students have access to computers or tablets, the survey may be administered on widely available online platforms such as Google Forms or Survey Monkey. The CMS results from such platforms are easily downloaded into spreadsheets for analysis. From spreadsheets, a simple graph generator can convert raw data into student-friendly bar graphs (a version of one graph generator is available from the authors). These graphs display simple frequency counts of the responses to each

item to ease students' understanding. The technology not only eliminates the need for manual scoring but improves data accuracy and facilitates seamless data conversion for easier interpretation. Furthermore, online drives such as SharePoint can provide secure storage for classroom data and meeting notes while facilitating easy sharing between team members.

### 7. Respects Classroom Culture

Ultimately, the refined classroom routines that emerge from the ClassMaps strategy reflect the culture of the class and the community. Both students and teachers 'own' the refined routines that emerged out of their shared conversations. While teachers retain their authority over their own teaching, the focus on students' experiences and their participation in action planning builds student ownership of classroom changes.

### 8. Summary

Because the ClassMaps strategy's action plans emerge out of an active collaboration of teachers with their students, the resulting classroom changes are more ecologically valid and more likely to persist over time. Because very substantial developmental risk and resilience science is systematically integrated into the ClassMaps Survey, the classroom collaboration focuses change on the protective factors of the classroom that reinforce students' academic success and their autonomy. The intent is to serve as one example of the charge that Miller [29] gave in his presidential address to the American Psychological Association: the ClassMaps strategy intends to give psychology away.

**Author Contributions:** The first draft of this manuscript was written by B.D. and both authors contributed to the final manuscript. All authors have read and agreed to the published version of the manuscript.

**Funding:** This manuscript received no external funding.

**Institutional Review Board Statement:** This is not a research article and the manuscript's preparation does not require ethical approval.

**Informed Consent Statement:** This is not a research article and the manuscript's preparation does not require an informed consent statement.

**Data Availability Statement:** This is not a research article and does not report on a set of research data.

**Conflicts of Interest:** The authors declare no conflicts of interest.

### Appendix A. The ClassMaps Survey

ClassMaps [2007] (Copyright Beth Doll; permission to copy is granted by the author)
Directions: These questions ask what is true about your class. For each question, circle the choice that is true for you. Do not put your name on the paper. No one will know what your answers are.

| I am a: | ☐ BOY/MALE | ☐ GIRL/FEMALE | I am in the _____ grade. |

**Believing in Me**

1.   I can do my work correctly in this class.

| NEVER | SOMETIMES | OFTEN | ALMOST ALWAYS |

2.   I can do as well as most kids in this class.

| NEVER | SOMETIMES | OFTEN | ALMOST ALWAYS |

3.  I can help other kids understand the work in this class.

    NEVER            SOMETIMES            OFTEN            ALMOST ALWAYS

4.  I can be a very good student in this class.

    NEVER            SOMETIMES            OFTEN            ALMOST ALWAYS

5.  I can do the hard work in this class.

    NEVER            SOMETIMES            OFTEN            ALMOST ALWAYS

6.  I can get good grades when I try hard in this class.

    NEVER            SOMETIMES            OFTEN            ALMOST ALWAYS

7.  I know that I will learn what is taught in this class.

    NEVER            SOMETIMES            OFTEN            ALMOST ALWAYS

8.  I expect to do very well when I work hard in this class.

    NEVER            SOMETIMES            OFTEN            ALMOST ALWAYS

**My Teacher**

9.  My teacher listens carefully to me when I talk.

    NEVER            SOMETIMES            OFTEN            ALMOST ALWAYS

10. My teacher helps me when I need help.

    NEVER            SOMETIMES            OFTEN            ALMOST ALWAYS

11. My teacher respects me.

    NEVER            SOMETIMES            OFTEN            ALMOST ALWAYS

12. My teacher likes having me in this class.

    NEVER            SOMETIMES            OFTEN            ALMOST ALWAYS

13. My teacher makes it fun to be in this class.

    NEVER            SOMETIMES            OFTEN            ALMOST ALWAYS

14. My teacher thinks I do a good job in this class.

    NEVER            SOMETIMES            OFTEN            ALMOST ALWAYS

15. My teacher is fair to me.

    NEVER            SOMETIMES            OFTEN            ALMOST ALWAYS

**Taking Charge**

16. I want to know more about the things we learn in this class.

    NEVER            SOMETIMES            OFTEN            ALMOST ALWAYS

17. In this class, I can guess what my grade will be when I turn in my work.

    NEVER            SOMETIMES            OFTEN            ALMOST ALWAYS

18. I work as hard as I can in this class.

    NEVER            SOMETIMES            OFTEN            ALMOST ALWAYS

19. I find and fix my mistakes before turning in my work.

    NEVER            SOMETIMES            OFTEN            ALMOST ALWAYS

20. I learn because I want to and not just because the teacher tells me to.

    NEVER            SOMETIMES            OFTEN            ALMOST ALWAYS

21. When the work is hard in this class, I keep trying until I figure it out.

    NEVER            SOMETIMES            OFTEN            ALMOST ALWAYS

22. I know the things I learn in this class will help me outside of school.

    NEVER            SOMETIMES            OFTEN            ALMOST ALWAYS

23. I can tell when I make a mistake on my work in this class.

    NEVER            SOMETIMES            OFTEN            ALMOST ALWAYS

**My Classmates**

24. I have a lot of fun with my friends in this class.

    NEVER            SOMETIMES            OFTEN            ALMOST ALWAYS

25. My friends care about me a lot.

    NEVER            SOMETIMES            OFTEN            ALMOST ALWAYS

26. I have friends to eat lunch with and play with at recess.

    NEVER            SOMETIMES            OFTEN            ALMOST ALWAYS

27. I have friends that like me the way I am.

    NEVER            SOMETIMES            OFTEN            ALMOST ALWAYS

28. My friends like me as much as they like other kids.

    NEVER            SOMETIMES            OFTEN            ALMOST ALWAYS

29. I have friends who will stick up for me if someone picks on me.

    NEVER            SOMETIMES            OFTEN            ALMOST ALWAYS

**Following the Class Rules**

30. Most kids work quietly and calmly in this class.

    NEVER            SOMETIMES            OFTEN            ALMOST ALWAYS

31. Most kids in this class listen carefully when the teacher gives directions.

    NEVER            SOMETIMES            OFTEN            ALMOST ALWAYS

32. Most kids follow the rules in this class.

    NEVER            SOMETIMES            OFTEN            ALMOST ALWAYS

33. Most kids in this class pay attention when they are supposed to.

    NEVER            SOMETIMES            OFTEN            ALMOST ALWAYS

34. Most kids do their work when they are supposed to in this class.

    NEVER            SOMETIMES            OFTEN            ALMOST ALWAYS

35. Most kids in this class behave well even when the teacher isn't watching.

    NEVER            SOMETIMES            OFTEN            ALMOST ALWAYS

**Talking With My Parents**

36. My parents and I talk about my grades in this class.

    NEVER            SOMETIMES            OFTEN            ALMOST ALWAYS

37. My parents and I talk about what I am learning in this class.

    NEVER            SOMETIMES            OFTEN            ALMOST ALWAYS

38. My parents and I talk about my homework in this class.

    NEVER            SOMETIMES            OFTEN            ALMOST ALWAYS

39. My parents help me with my homework when I need it.

    NEVER            SOMETIMES            OFTEN            ALMOST ALWAYS

40. My parents and I talk about ways that I can do well in school.

    NEVER            SOMETIMES            OFTEN            ALMOST ALWAYS

41. My parents and I talk about good things I have done in this class

    NEVER        SOMETIMES        OFTEN        ALMOST ALWAYS

42. My parents and I talk about problems I have in this class.

    NEVER        SOMETIMES        OFTEN        ALMOST ALWAYS

**I worry that . . ..**

43. I worry that other kids will do mean things to me.

    NEVER        SOMETIMES        OFTEN        ALMOST ALWAYS

44. I worry that other kids will tell lies about me.

    NEVER        SOMETIMES        OFTEN        ALMOST ALWAYS

45. I worry that other kids will hurt me on purpose.

    NEVER        SOMETIMES        OFTEN        ALMOST ALWAYS

46. I worry that other kids will say mean things about me.

    NEVER        SOMETIMES        OFTEN        ALMOST ALWAYS

47. I worry that other kids will leave me out on purpose.

    NEVER        SOMETIMES        OFTEN        ALMOST ALWAYS

48. I worry that other kids will try to make my friends stop liking me.

    NEVER        SOMETIMES        OFTEN        ALMOST ALWAYS

49. I worry that other kids will make me do things I don't want to do.

    NEVER        SOMETIMES        OFTEN        ALMOST ALWAYS

50. I worry that other kids will take things away from me.

    NEVER        SOMETIMES        OFTEN        ALMOST ALWAYS

**Kids In This Class**

51. Kids in this class argue a lot with each other.

    NEVER        SOMETIMES        OFTEN        ALMOST ALWAYS

52. Kids in this class pick on or make fun of each other.

    NEVER        SOMETIMES        OFTEN        ALMOST ALWAYS

53. Kids in this class tease each other or call each other names.

    NEVER        SOMETIMES        OFTEN        ALMOST ALWAYS

54. Kids in this class hit or push each other.

    NEVER        SOMETIMES        OFTEN        ALMOST ALWAYS

55. Kids in this class say bad things about each other.

    NEVER        SOMETIMES        OFTEN        ALMOST ALWAYS

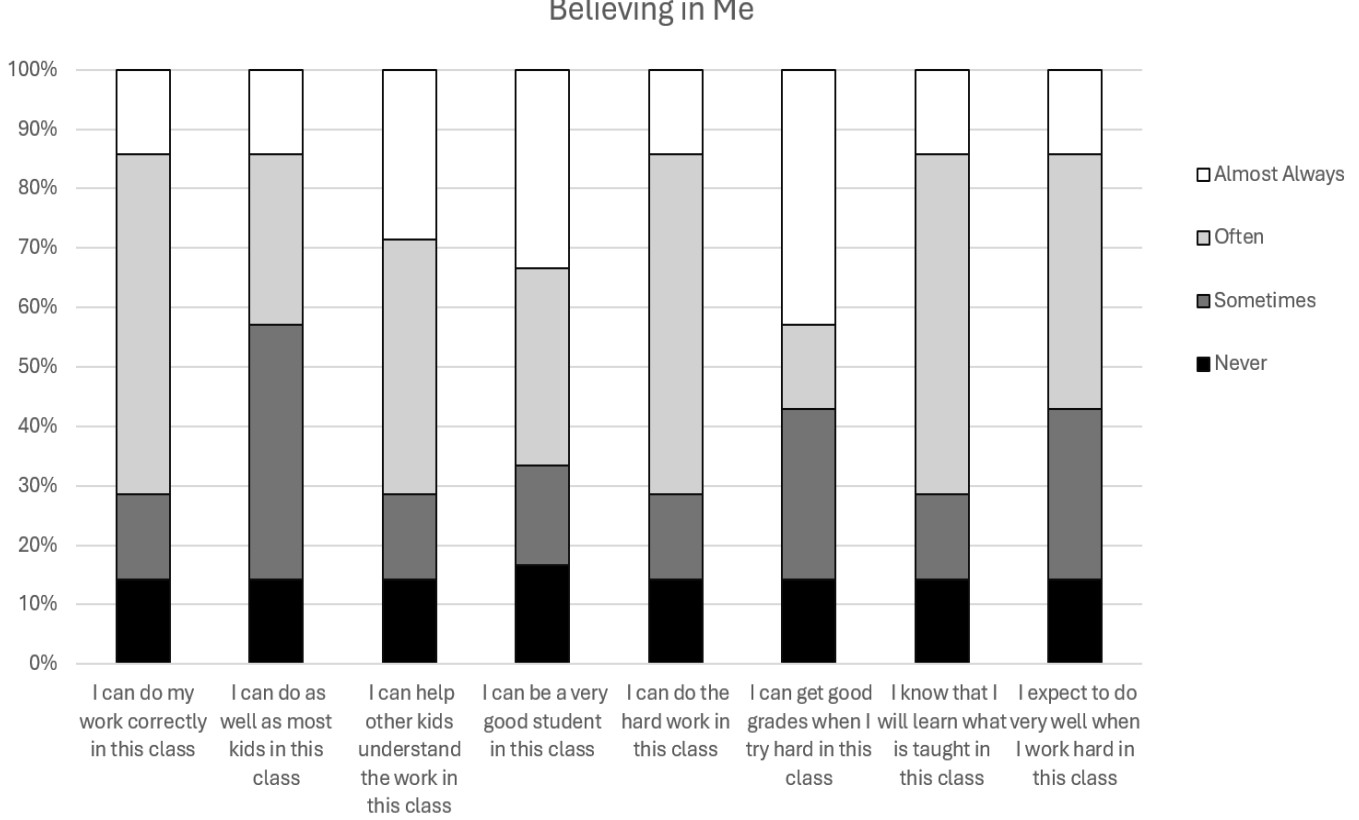

**Figure A1.** Sample graph.

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
