# Peer review of "Building Resilience into Classrooms: A Participatory Action Approach"

_education, doi:10.3390/educsci14050511_

Round 1
Reviewer 1 Report
Comments and Suggestions for Authors
The article provides a useful overview of the CMS as a tool to positively influence class and school culture. I would suggest adding, in the abstract and introduction, the age range for which it has been found to be effective. The article is well written in correct English. References are appropriate.
Author Response
Thank you for this very useful suggestion; I have added age ranges to the manuscript in two place.
Reviewer 2 Report
Comments and Suggestions for Authors
The article is well conceived in terms of subject matter, literature review and some of its conclusions. But its scientific structure is somewhat incomprehensible. Instead of using an IMRD scheme, the article has a succession of sections that mix empirical results. On the one hand (341-381), it talks about research that looks like field research itself, but its evidence is not clear because the sample is not well detailed and, more importantly, the results are not shown graphically and clearly. On the other hand (297-310), there are results that look like bibliometric research, but this is not clearly detailed either. Although an expert, in-depth look can distinguish the quality of the research, I think it is essential that the authors reconstruct and place the elements of the article with a more traditional and conventional IMRD structure and with all the methodological part well detailed and illustrate, in order to facilitate the reading of the article. After this first step, we will be able to make a more precise revision of some aspects that do not seem clear, although perhaps they will become clearer with this new configuration of the data, the analyses, the discussion and the conclusions.
Author Response
I appreciate the reviewer's comments that the manuscript was well referenced and supported. I note that the special issue sought contributions that can include "conceptual and theoretical discussions" and "reviews and meta-analyses" and was not restricted to empirical studies. This paper was a conceptual and theoretical review and so is not appropriate for an IMRD organization.
Round 2
Reviewer 2 Report
Comments and Suggestions for Authors
I do not understand authors claim that "this paper was a conceptual and theoretical review". The article has numerous descriptions on a real experience. For instance: "First, trial and error was used 162
to compare and contrast three alternative measurement strategies in middle schools (6th 163
to 8th grades): direct observation, teacher reports, and student reports of each school char- 164
acteristic." and many more
Either this intervention data are more clearly described (and IMRD format needed) either they are eliminated to leave only the theoretical review.
Author Response
I understand now what this reviewer's concern is and I have reworded this section of the paper to be clear that this early comparison of alternative strategies for assessing classroom protective factors was not an empirical study.